# Newly Synthesized Morpholinyl Mannich Bases as Corrosion Inhibitors for N80 Steel in Acid Environment

**DOI:** 10.3390/ma15124218

**Published:** 2022-06-14

**Authors:** Yuhao Chen, Zhonghua Chen, Yaowen Zhuo

**Affiliations:** 1College of Materials Science and Engineering, South China University of Technology, Guangzhou 510641, China; 18804630341@163.com; 2Yueyang Chem Waterborne Additive Co., Ltd., Yueyang 414000, China; zhuoyaowen@chem-additive.com

**Keywords:** N80 steel, acid corrosion, Mannich bases, inhibitor, electrochemical tests

## Abstract

New Mannich bases, 3-morpholino-1-phenylpropan-1-one (MPO) and 3-morpholino-1-phenyl-3-(pyridin-4-yl) propan-1-one (MPPO), were synthesized, characterized, and studied as corrosion inhibitors for N80 steel in 1 M hydrochloric acid (HCl) solution using weight loss, potentiodynamic polarization, electrochemical impedance spectroscopy, scanning electron microscopy, X-ray photoelectron spectroscopy, and FT-IR spectroscopy. The inhibition efficiency increases with increasing inhibitor concentrations, and the corrosion inhibition efficiency of the MPO and MPPO could reach 90.3% and 91.4%, respectively, at a concentration of 300 ppm at 305 K. The effect of the temperature on the corrosion inhibition behavior of inhibitors was discussed. Electrochemical tests showed that the synthesized inhibitors are mixed. The EIS test results showed that the presence of MPO and MPPO reduced the double-layer capacitance in the corrosion process, thereby reducing the charge transfer resistance. The SEM and EDX results showed that the MPO and MPPO formed a uniform adsorption film on the surface of the N80 steel. The adsorption mechanism of the inhibitors was simulated with different adsorption models and the results showed that the inhibitors were the chemisorbed type. The results of the FT-IR spectroscopy proved that the inhibitor interacted with metal atoms on the steel surface.

## 1. Introduction

Corrosion protection of steel and its alloys is crucial in numerous fields of industry. Particularly in the petroleum industry, acidification is an effective technique to extend productivity. However, acid corrosion of oil wells is a crucial engineering problem due to the utilization of high concentrations of acid [1,2,3]. To reduce the corrosion hazards of metal pipes and metal tanks, it is necessary to use corrosion inhibitors to inhibit the corrosion of acidic solutions [4]. Corrosion inhibitors include inorganic inhibitors and organic inhibitors. Inorganic inhibitors usually include chromates, dichromates, tetraborates, molybdates, arsenates, nitrites, and phosphates [5]. These inorganic compounds such as chromates and nitrites are restricted to some extent due to their high biological toxicity, environmental pollution, and high cost. Therefore, organic corrosion inhibitors are one of the most promising methods to mitigate corrosion. Most organic corrosion inhibitors chiefly contain O, N, and S heteroatoms, aromatic rings, and multi-bonds, all of which can contribute to the lone pairs of electrons, allowing for stronger adsorption of the inhibitor molecule to the metal surface [6,7,8]. It can physically or chemically interact with the metal surface and adsorb firmly on the metal surface, thus forming a protective film that can isolate the corrosive medium [9].

Over the past few decades, several studies have been reported on the corrosion inhibition properties of organic corrosion inhibitors on mild steels under acidic conditions, including Mannich base [10,11], schiffine [12,13], pyridine [14,15], imidazole [6], quinoline [16,17], etc. Mannich base compounds have attracted abundant attention because of their aromatic rings, heteroatom structures, and good acid solubility. The existence of Π electrons of aromatic rings and lone electron pairs of heteroatoms makes Mannich base compounds adsorb on metal surfaces efficiently and decreases the metal corrosion rate. Jeeva et al. [18] investigated the effect of nicotinamide and its Mannich derivatives on mild steel in HCl solution. Recent studies have reported that morpholine Mannich base and its derivatives have an excellent corrosion inhibition effect on low carbon steel in an acid solution [19]. Lavanya et al. [20] studied the corrosion inhibition effect of morpholine thiourea new Mannich base on mild steel in an HCl solution. Nanji et al. [21] reported morpholine derivatives as corrosion inhibitors for mild steel in an HCl medium.

In this paper, Mannich base corrosion inhibitor 3-morpholino-1-phenylpropan-1-one (MPO) and 3-morpholin-1-phenyl-3- (pyridin-4-yl) propane-1-one (MPPO) was synthesized from morpholine, pyridine-4-formaldehyde, and acetophenone. The corrosion inhibitor has multiple active adsorption centers for O, N heteroatoms and conjugated bonds containing lone electron pairs, and the hydrophobic alkyl chains connecting different adsorption structures can isolate the metal surface from the corrosion medium. The corrosion inhibition performance of MPO and MPPO on N80 steel in 1M HCl solution was studied using the weight-loss method, electrochemical analysis, and surface analysis.

## 2. Materials and Methods

### 2.1. Synthesis of Mannich Base

Morpholine, pyridine-4-formaldehyde, 37% formaldehyde solution, acetophenone, 37% hydrochloric acid, and ethanol: AR, Shanghai Aladdin Biochemical Technology Co., Ltd., (Shanghai, China). The synthetic route of Mannich bases is shown in Figure 1. In general, morpholine (0.5 mol) was dissolved in 30 mL anhydrous ethanol solution, adjusted to pH 2–3 with a concentrated HCl solution, then acetophenone (0.5 mol) was added, and then slowly dropped into the aldehyde (0.5 mol) (formaldehyde and pyridine-4-formaldehyde). The mixture was stirred at 85 °C for 10 h and then cooled to room temperature. After standing overnight, the sediment is filtered. The crude product was washed three times with cold ethanol and then recrystallized with anhydrous ethanol to obtain a pure compound [22,23]. The melting points and yields of the products were determined, and the structures of the products were characterized by FT-IR and NMR.

*3-morpholino-1-phenylpropan-1-one (MPO)**:* White solid (85%, yield); mp 143–146 °C; IR (KBr, cm^−1^): 2981, 2859, 2459, 1687, 1597, 1454, 1223, 1128, 1089 750, 692. ^1^H NMR (600 MHz, CDCl3): δ(ppm) 2.39(s, 4H), 3.47(m, 4H), 3.76(m, 2H), 3.93(m, 2H), 7.42(m, 2H), 7.55(m, 1H), 7.93(m, 2H). ^13^C NMR (600 MHz; CDCl_3_): δ(ppm) 32.36, 51.96, 52.21, 63.63, 128.15, 128.93, 134.48, 135.21, 199.43.

*3-morpholino-1-phenyl-3-(pyridin-4-yl) propan-1-one (MPPO)**:* White solid (89%, yield); mp 147–150 °C; IR (KBr, cm^−1^): 2963, 2811, 1695, 1593, 1455, 1273, 1120, 1026, 762, 680. ^1^H NMR (600 MHz, CDCl3): δ(ppm) 2.35(s, 4H), 3.51–3.53(m, 4H), 3.63(m), 7.12–7.13(m, 1H), 7.42–7.44(m, 2H), 7.58(m, 1H), 7.85(m, 2H), 8.04(m, 2H), 8.65–8.67(m, 2H). ^13^C NMR (600 MHz, CDCl_3_): δ(ppm) 46.43, 49.56, 67.02, 68.12, 127.79, 128.84, 129.76, 131.61, 134.09, 134.47, 197.33.

### 2.2. Material Preparation

The N80 steel specimens (50.00 mm × 10.00 mm × 3.00 mm: length, width, height) are purchased from Baosteel Co., Ltd., (Shanghai, China), with the chemical composition (wt.%) of C 0.42, Si 0.23, Mn 1.68, P 0.012, S 0.011, Cr 0.02, Ni 0.01, Mo 0.18, Cu 0.03, and Fe the rest. The surface of N80 steel specimens was ground with 800# to 1200# (# represents mesh) sanding paper, then washed with ethanol and distilled water, dried, and used for experimental testing.

### 2.3. Weight Loss Measurements

The N80 steel specimens were immersed in 1 M HCl solutions containing different concentrations of inhibitors at different temperatures for 2 h. The mass loss before and after immersion was determined using an analytical balance (HZT-B2000, Fuzhou Huazhi Scientific Instruments Co., Fuzhou, China). The temperature of the solution in the experiment was controlled by a thermostat. Each set of experiments was repeated three times in all identical cases and the results were averaged. The corrosion rate (C_R_) of the steel plates was obtained from the following equation:(1)$$C_{R}=\frac{\left(W_{0}-W_{i}\right)}{A\times T}$$
where A is the surface area of the steel specimen (cm^2^) and T is the immersion time (h). The inhibition efficiency and coverage are calculated by the following formula:(2)$$\mathrm{IE}\left(\%\right)=\frac{W_{0}-W_{i}}{W_{0}}\times 100$$
(3)$$\Theta =\frac{W_{0}-W_{i}}{W_{0}}$$
where W_0_ and W_i_ are the values of the weight loss without and with the addition of the inhibitor, respectively [24,25].

### 2.4. Electrochemical Measurements

The electrochemical experiments were carried out on a CS310 electrochemical workstation from Wuhan COST Instruments Co., Ltd., (Wuhan, China). At a constant temperature of 305 K, 1 M HCl was used as the electrolyte, N80 steel specimen as the working electrode (WE), platinum gauze electrode as the auxiliary electrode, and saturated calomel electrode (SCE) as the reference electrode (RE) [26]. The N80 steel specimens after grinding were immersed in 100 mL 1M HCl solution containing different concentrations of inhibitors for 1 h to obtain a stable state. Then the values of open circuit potential (OCP) were tested and recorded, and EIS and Tafel polarization experiments were performed. Tafel curves were obtained by changing the electrode potential automatically from −250 to +250 mV versus OCP at a scan rate of 10 mV s^−1^.The electrochemical impedance test was carried out in the frequency range of 0.1~10,000 Hz by applying 0.01 V sinusoidal AC voltage [27].

### 2.5. Surface Analyses

The N80 steel specimens of size were immersed in 1 M HCl solution without and with a corrosion inhibitor concentration of 300 ppm of MPO and MPPO for 2 h. The immersed steel specimens were washed with ethanol and distilled water, dried and sheared to a size of 1 cm^2^ for SEM and EDX characterization. Observations were performed on Evo 18 XVP instrument, Zeiss, Carl Zeiss AG, Oberkochen, Germany, at an accelerating voltage of 20 kV and a magnification of 2K times. To study the interaction between the corrosion inhibitor and the surface of the N80 steel, the solid products on the steel surface of the weight loss experiment were scraped off for FT-IR spectroscopic study (VERTEX 70, Bruker, Germany).

## 3. Results

### 3.1. Weight Loss Measurements

#### 3.1.1. Concentration Effect of MPO/MPPO on Corrosion Inhibition

The weight loss data for the N80 steel specimens in the absence and presence of different concentrations of inhibitors in 1 M HCl over a temperature range of 305–335 K are listed in Table 1. According to the data, the corrosion rate of the synthesized inhibitor decreases rapidly with increasing concentration. It is observed that all inhibitors showed very high efficiency at the optimum concentration of 300 ppm, and there is no remarkable increase in the inhibition efficiency for a higher concentration of inhibitors. The C_R_ values of the MPO and MPPO were only 0.19 mg cm^−2^h^−1^ and 0.16 mg cm^−2^h^−1^, respectively, at a concentration of 300 ppm at 305 K. However, in the sample without an inhibitor, the C_R_ value was 1.86 mg cm^−2^h^−1^. In addition, the coverage and corrosion inhibition efficiency of the MPO and MPPO gradually increased with the increase in inhibitor concentration, and the highest the inhibition efficiency could reach was 89.8% and 91.4%, respectively. This indicates that as the inhibitor concentration increases, more of the inhibitor adsorbs on the N80 steel surface, thus isolating the steel surface from the corrosive medium.

#### 3.1.2. Temperature Effect of MPO/MPPO on Corrosion Inhibition

To investigate the effect of temperature on the corrosion process of N80 steel in 1M HCl solution, the weight loss experiments were carried out at a temperature range of 305 K–335 K in the absence and presence of different concentrations of inhibitors during 2 h immersion. The effect of temperature on corrosion rate is shown in Table 1. It is clear that the corrosion rate increases with the increasing solution temperature for both the presence and absence of inhibitors. The increase in temperature may lead to the desorption of the inhibitor on the N80 steel surface. However, it can be seen from Table 1 that as the temperature increases, the greater the increase in corrosion rate without the inhibitor compared to the presence of the inhibitor. In the absence of inhibitors, the corrosion rate is only 1.86 mg cm^−2^h^−1^ at 305 K, whereas at 335, K is 3.76 mg cm^−2^h^−1^, showing a dramatic increase in corrosion rate with the increasing solution temperature. In the presence of inhibitors, the corrosion rate increases slowly with the temperature of the solution, from 0.19 mg cm^−2^h^−1^ at 305 K to 1.08 mg cm^−2^h^−1^ at 335 K for the MPO, and from 0.16 mg cm^−2^h^−1^ to 1.03 mg cm^−2^h^−1^ for the MPPO at 300 ppm. The prosperous corrosion inhibition performance results from the inhibitors forming a protective film on the steel surface.

A relationship between the corrosion rate (C_R_) and absolute temperature (T) can be expressed by the Arrhenius equation [28,29].
(4)$$C_{R}={\mathrm{Ae}}^{\frac{-E_{a}}{\mathrm{RT}}}$$
where A is the pre-exponential factor, E_a_ is the apparent activation energy, R is the universal gas constant, and T is the absolute temperature.

The relationship between the logarithm of the corrosion rate and 1/T obtained from Table 1 is shown in Figure 2. The values of activation energy obtained from the slope (−E_a_/2.303R) of the straight lines are given in Table 2.

The enthalpy of activation ($-∆H_{\mathrm{corr}}^{*})\;$ and entropy of activation ($∆S_{\mathrm{corr}}^{*}$) for the N80 steel dissolution process are obtained from the Eyring transition state equation [28,30].
(5)$$C_{R}=\frac{\mathrm{RT}}{N_{A}h}\exp\frac{∆S_{\mathrm{corr}}^{*}}{R}\exp\frac{-∆H_{\mathrm{corr}}^{*}}{\mathrm{RT}}$$
where h is the Planck’s constant and N_A_ is the Avogadro number. A plot of ln(C_R_/T) and 1/T gave a straight line as shown in Figure 2 with a slope ($-∆H_{\mathrm{corr}}^{*}$/2.303R) and intercept [ln(R/N_A_h)+$∆S_{\mathrm{corr}}^{*}$/2.303R]. The values of enthalpy and entropy of activation were calculated by these plots and are given in Table 2.

The values of E_a_ increase with the increasing concentration of inhibitors, indicating the inhibitors act as a barrier in metal corrosion reactions. The groups with inhibitors have a higher activation energy value compared to the group without inhibitors. This is due to the occurrence of inhibitors in the adsorption process on the metal surface, leading to the formation of a physical protective layer on the metal surface, which blocks the corrosion charge transfer. The positive value of $∆H_{\mathrm{corr}}^{*}$ indicates that the corrosion of the N80 steel in 1 M HCl is an endothermic process, and it presents higher in the presence of inhibitors than in the absence of inhibitors. It is also shown that the values of E_a_ and $∆H_{\mathrm{corr}}^{*}$ enlarge with the increasing concentration of inhibitors, indicating that the Mannich base inhibitors amplify the energy barrier to the dissolution of N80 steel in the HCl solution, leading to a suppressive role for N80 steel in the corrosion reaction. Identically, the value of entropy of activation in the presence of inhibitors is negative and larger, showing that the activated complex in the rate-determining step is associative and the activated complex has a more orderly structure [30]. Simultaneously, the values of $∆S_{\mathrm{corr}}^{*}$ enhance with inhibitor concentration suggesting that MPO and MPPO play an excellent role in hindering the corrosion process.

### 3.2. Electrochemical Corrosion Measurements

Figure 3 shows the steady-state OCP curves of N80 steel specimens in 1 M HCl solutions containing different concentrations of inhibitors. Figure 4 shows the cathodic and anodic polarization curves of N80 steel after 1 h immersion in 1 M HCl without and with various concentrations of MPO and MPPO at 305 K, respectively. The polarization curves of both cathode and anode shift to the lower current densities with the addition of MPO and MPPO, indicating that the inhibitor molecules suppressed the corrosion of steel in 1.0 M HCl. Table 3 shows the numerical values of electrochemical corrosion kinetic parameters such as corrosion current density (I_corr_), corrosion potential (E_corr_), cathodic and anodic Tafel slops (β_a_, β_c_), and the inhibition efficiency (IE).

The inhibition efficiency (IE, %) from potentiodynamic polarization was calculated using the following equation [31]:(6)$$\mathrm{IE}\%=\frac{I_{0}-I^{\prime }}{I_{0}}\times 100$$
where I_0_ and I′ are the corrosion current density without and with the inhibitor, respectively.

It is obvious from Table 3 that the numerical values of I_corr_ noticeably decrease with the addition of MPO and MPPO in 1.0 M HCl. Moreover, the values of I_corr_ gradually decrease with the concentration of inhibitors due to more inhibitor molecules absorbing on the steel surface. Furthermore, there is no significant change in the corrosion potential with the addition of MPO compared to the sample without the corrosion inhibitor, while the corrosion potential shifted in a more negative direction with the addition of MPPO., Besides this, the maximum E_corr_ in the presence of inhibitors shifts less than ±85 mV from that of the uninhibited solution. These illustrate that MPO and MPPO are mixed inhibitors and retard the anodic dissolution and subsequent reduction of the H_2_ evolution reaction [32]. In brief, the two inhibitors studied showed a high inhibition efficiency, which increased with the increasing inhibitor concentration, mainly due to the adsorption of the inhibitor on the steel surface and thus the formation of a protective film.

### 3.3. Electrochemical Impedance Spectroscopy (EIS)

The effect of MPO and MPPO corrosion inhibitors on the corrosion behavior of N80 steel specimens in 1M HCl solution was investigated by EIS test after 1 h immersion at 305 K. Figure 5 shows Nyquist and Bode plots without and with various concentrations of MPO and MPPO. It is shown in Figure 5 that the Nyquist plots in the presence of MPO and MPPO are characterized by a large semicircle with the center located below the real axis. The addition of MPO and MPPO does not affect the shape of the semicircular arc, which indicates that the presence of an inhibitor does not change the corrosion mechanism of N80 steel in 1M HCl. Whereas, the diameter of these semicircles significantly increases with the increasing concentration of inhibitors, which means that more corrosion inhibitor molecules adhere to the metal surface. The corresponding Bode plots were shown in Figure 5c,d, and generally, the logarithm of resistance in the low-frequency region represents the ability to block charge transfer. The logarithm of resistances of the MPO and MPPO prominently increases with the concentrations of both inhibitors.

Furthermore, the constructed equivalent circuit model is shown in Figure 6. In the equivalent circuit model, R_s_ is solution resistance, R_f_ is film resistance, R_CT_ is the charge transfer resistance, R_p_ is the sum of R_f_ and R_CT_, CPE_f_ is the constant phase element of film and CPE_dl_ is the constant phase element of the double-layer [32]. The impedance of CPE (Z_CPE_) can be given as follows [33]:(7)$$Z_{\mathrm{CPE}}=Y_{0}^{-1}{\left(j\omega \right)}^{-n}$$
where Y_0_ is the amplitude of CPE, and n is the phase shift. The n value represents the deviation of the inhibitor from the ideal performance, and the value ranges between 0 and 1. X is the angular frequency represented by rad s^−1^ and j is an imaginary number. The effective capacitance values were calculated as follows [34]:(8)$$C_{\mathrm{dl}}=Y_{0,\mathrm{dl}}^{\frac{1}{n}}{\left(\frac{1}{R_{s}}+\frac{1}{R_{\mathrm{CT}}}\right)}^{\frac{n-1}{n}}$$
(9)$$C_{f}=Y_{0,\mathrm{dl}}^{\frac{1}{n}}\times R_{f}^{\frac{1-n}{n}}$$

The electrochemical parameters are given in Table 4. The values of Y_0,dl_ and C_dl_ decrease with the addition of the MPO and the MPPO, which is attributed to the increase in the thickness of the protective film on the electrode surface. Furthermore, the values of charge transfer resistance (R_CT_) significantly increase with the presence of MPO and MPPO. The inhibition efficiency value was calculated using the following equation.
(10)$$\mathrm{IE}\left(\%\right)=\frac{R_{p}-R_{p}^{0}}{R_{p}}\times 100$$
where Rp and $R_{p}^{0}$ are the polarization resistance values in the presence and absence of the three inhibitors, respectively [35,36]. The EIS data show that the MPO and MPPO show good performance in corrosion inhibition, with an inhibition efficiency of up to 91.31% and 91.48% at 300 ppm, respectively, and the MPPO has a higher inhibition efficiency than the MPO.

Compared with other Mannich base corrosion inhibitors in the literature, such as the bis-Mannich base studied by Zhang et al. [32] and the imidazoline-based Mannich base synthesized and studied by Zhu et al. [37] for corrosion inhibition of N80 steel in HCl solution, the morpholine-based Mannich base corrosion inhibitor synthesized in this paper still has a high inhibition efficiency in a high concentration of acid.

### 3.4. Adsorption Isotherm

Generally, the corrosion of metals immersed in an acid solution is principally due to the presence of water molecules and acid medium on the metal surface. The robust absorption of organic inhibitor molecules on the metal surface through the polar amine group and benzene ring replaces the H_2_O molecules at the corroding metal/solution interface. This process can be described by the following reaction scheme [38]:Org_soln_ + xH_2_O_ads_ ↔ Org_ads_ + xH_2_O_soln_
where the x is the number of water molecules displaced by one molecule of organic inhibitor. Basic information on the interaction between inhibitors molecules and metal surfaces can be obtained by adsorption isotherms. Data from weight loss experiments were used to fit the isotherms of Langmuir and Dubinin–Radushkevich. The linear regression fitting to the Langmuir adsorption isotherm is illustrated by plotting C_inh_/θ versus C_inh_ (Figure 7) according to the following equation [39]:(11)$$\frac{C_{\mathrm{inh}}}{\theta }=\frac{1}{K_{\mathrm{ads}}}+C_{\mathrm{inh}}$$
where the C_inh_ is the concentration of inhibitor, K_ads_ is the adsorption equilibrium constant, and θ is the degree of surface coverage. The linear regression coefficient (R^2^) obtained from the data of regression fitting is close to 1.0, indicating that the Langmuir isotherm is suitable for modeling the adsorption of MPO and MPPO on metal surfaces.

The values of K_ads_ were calculated by the intercept of Langmuir isotherm plots and are listed in Table 5. The values of Gibb’s standard free energy of adsorption ($∆G_{\mathrm{ads}}^{0}$) can be calculated from the following equation:(12)$$∆G_{\mathrm{ads}}^{0}=-\mathrm{RTln}55.5K_{\mathrm{ads}}$$
where R is the universal gas constant, T is the absolute temperature, and 55.5 is the molar concentration of water in the solution expressed in M (mol L^−1^). The values of $∆G_{\mathrm{ads}}^{0}$ are listed in Table 5. From the data, the K_ads_ values of the MPO and MPPO are relatively large, indicating that the inhibitor molecules are robustly absorbed on the steel surface and therefore improve the inhibition efficiency [28]. Moreover, the K_ads_ values decrease with the increasing solution temperature due to the desorption of the inhibitor molecules on the metal surface. Additionally, the $∆G_{\mathrm{ads}}^{0}$ values range between −32.7 and −36.1 kJ mol^−1^, which are lower than −20 kJ mol^−1^. This suggests that the absorption of the MPO and MPPO molecules on the steel involves both physisorption and chemisorption, but the latter is predominant [40,41].

Further, the experimental results were simulated using the Dubinin–Radushkevich isotherm model as the following equation [38,42]:(13)$$\ln\theta ={\ln\theta }_{\max}-{a\delta }^{2}$$
where the $\theta_{\max}$ is the maximum surface coverage, a is constant, and δ is the Polanyi potential, which can be given as:(14)$$\delta =\mathrm{RTln}\left(1+\frac{1}{C_{\mathrm{inh}}}\right)$$

The constant a represents the average adsorption energy (E_ads_), that is, the transfer energy of 1 mol of adsorbate from infinity (acidic solution containing inhibitor molecules) to the adsorbent surface (steel surface).
(15)$$E_{\mathrm{ads}}=\frac{1}{\sqrt{2a}}$$

From the studies of Noor and Solomon, if the values of E_ads_ are less than 8.0 kJ mol^−1^, the adsorption process belongs to physical adsorption, otherwise, it is chemical adsorption [42,43]. The values of E_ads_ (Table 5) obtained from the data confirm the chemical absorption of the MPO and MPPO molecules onto the N80 steel surface.

### 3.5. Surface Analysis

#### 3.5.1. SEM and EDX Analysis

The SEM micromorphology and corresponding EDS analysis of N80 steel after 2 h immersion in 1 M HCl solution in the absence and presence of inhibitors are shown in Figure 8. As shown in Figure 8 A, the steel surface was acutely eroded and highly damaged, and corrosion products and pit were widely distributed on it. The relevant EDS spectra and data showed comparatively high contents of oxygen and chlorine, indicating the uninhibited steel surface in the presence of many acid corrosion products. In contrast, the surfaces in the presence of inhibitors show smoother and lighter corrosion damage and the surfaces of the MPPO (Figure 8C) are smoother and more complete. Furthermore, the corresponding EDS spectra and data showed the contents of acid corrosion products were lower than the uninhibited group. Besides this, the EDS spectra showed a characteristic peak of nitrogen around the 0.3 kev, which verified that the inhibitors are successfully adsorbed on the surface of the N80 steel, forming a uniform and dense protective film. According to the results of the surface analysis, the inhibition effect of the MPPO is significantly higher than that of the MPO. This is due to the MPPO molecules having more heteroatoms and aromatic rings that contain more lone electron pairs. Therefore, the MPPO can be effectively adsorbed on the steel surface to form a protective film.

#### 3.5.2. FT-IR Analysis

After immersion in a 1 M hydrochloric acid solution of corrosion inhibitor at a concentration of 300 ppm for 2 h, the corrosion inhibitor film adsorbed on the surface of N80 steel was scraped off, collected, and then studied by infrared spectroscopy. The FT-IR spectra of the MPO, MPPO and the scraped samples of MPO-Fe and MPPO-Fe are shown in Figure 9. The comparison of the IR characteristic absorption frequencies of the scraped and pure samples is shown in Table 6. The C=O characteristic band frequencies of the MPO and MPPO shift to lower frequencies, 1687 to 1668 and 1695 to 1664, respectively, indicating a strong interaction of carbonyl oxygen with the vacant d-orbitals of the metal surface. The C=C peaks on the benzene ring as well as the C=N peaks on the pyridine ring have lower frequencies, 1597 to 1577 and 1593 to 1570, indicating that the benzene and pyridine rings interact with the metal surface. The stretching frequencies of C-O-C and C-N-C in the morpholine structure are shifted to lower frequencies, indicating that the O and N atoms in the morpholine molecule are also involved in the adsorption process. In addition, the appearance of characteristic frequencies of Fe-O and Fe-N in the specified regions indicates the existence of a coordination bond between the inhibitor molecule and the vacant d-orbitals of the metal atoms. The stretching bands of C=O, C=C, C=N, C-O-C, and C-N-C of the MPO and MPPO attached to the N80 surface were displaced, indicating that both corrosion inhibitors were successfully adsorbed on the N80 steel surface.

## 4. Conclusions

The adsorption and corrosion behavior of two morpholine Mannich inhibitors on N80 steel in 1.0 M HCl solution were systematically studied. The inhibition efficiency increased with the increase in inhibitor concentrations, with the corrosion inhibition efficiency reaching 90.3% and 91.4%, respectively, at a concentration of 300 ppm at 305 K. The values of E_a_ and $∆H_{\mathrm{corr}}^{*}$ increased with the increase in inhibitor concentration, indicating that the energy barrier of the corrosion reaction increased with the existence of the inhibitor. Electrochemical measurement results show that the two inhibitors are mixed inhibitors, which play a role in inhibiting anodic dissolution and cathodic hydrogen evolution reaction. SEM, EDX, and FT-IR spectra results showed that the corrosion inhibitor adsorbed on the surface of N80 steel and formed a dense protective film. The adsorption of MPO and MPPO on the N80 steel surface conformed to the Langmuir adsorption isotherm and the Dubinin–Radushkevich adsorption isotherm and confirmed that the adsorption type of these inhibitors was chemical adsorption. The theoretical calculation results are in good agreement with the experimental results, and the inhibition efficiency follows the order of MPO < MPPO.

## Figures and Tables

**Figure 1 materials-15-04218-f001:**
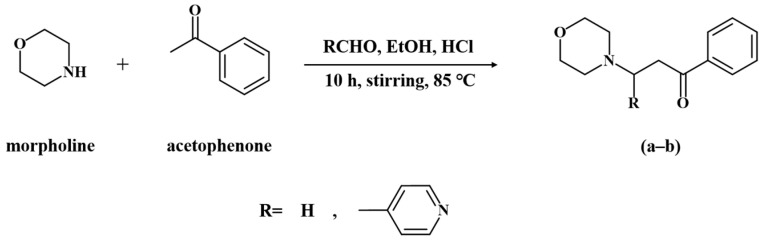
Scheme for synthesizing morpholinyl Mannich bases: (**a**) MPO and (**b**) MPPO.

**Figure 2 materials-15-04218-f002:**
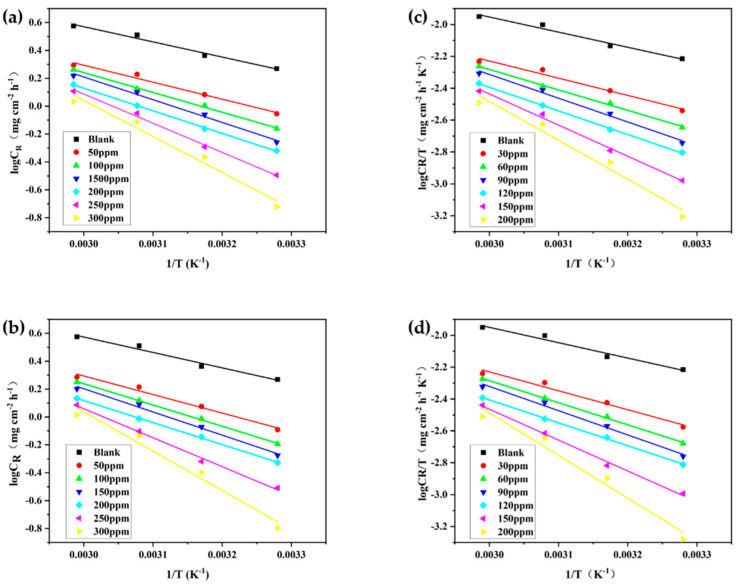
Arrhenius (**a**) MPO, (**b**) MPPO and Eyring (**c**) MPO, (**d**) MPPO plots for corrosion of N80 steel in 1 M HCl containing various concentrations of inhibitors.

**Figure 3 materials-15-04218-f003:**
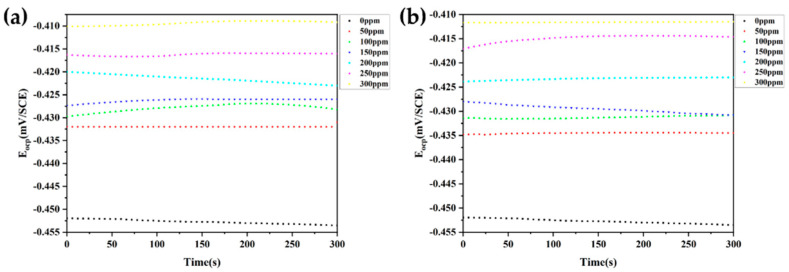
The variation of OCP with time for N80 steel in 1 M HCl solution containing different concentrations of (**a**) MPO, (**b**) MPPO at 305 K.

**Figure 4 materials-15-04218-f004:**
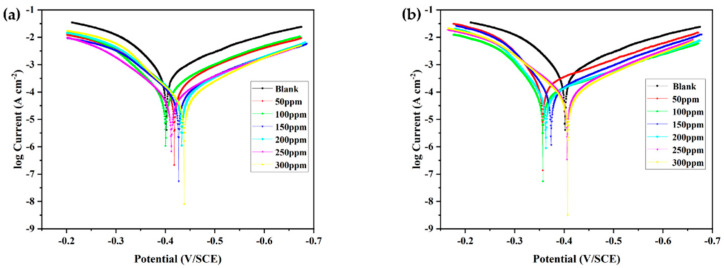
Polarization curves for N80 steel measured in uninhibited 1.0 M HCl and with different concentrations of inhibitor: (**a**) MPO and (**b**) MPPO.

**Figure 5 materials-15-04218-f005:**
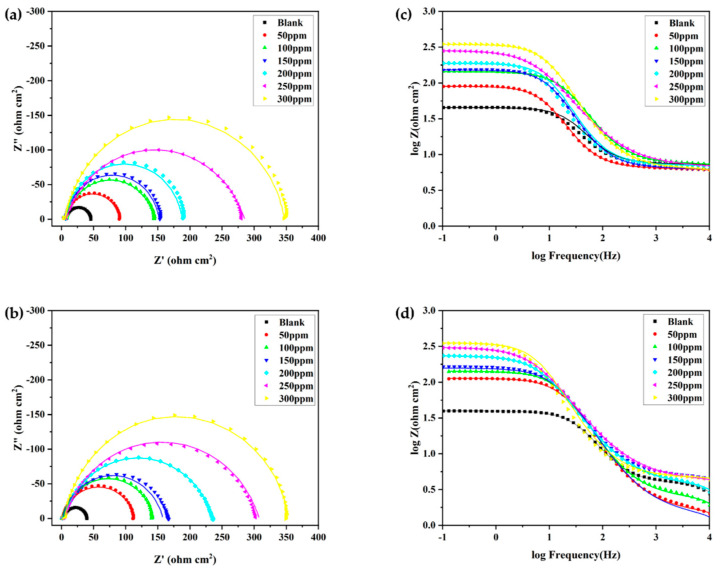
Nyquist (**a**) MPO, (**b**) MPPO and Bode (**c**) MPO, (**d**) MPPO plots of EIS measurements of N80 steel in 1 M HCl without and with different concentrations of inhibitors (characteristic symbol and real line represent experimental results and fitting results, respectively).

**Figure 6 materials-15-04218-f006:**
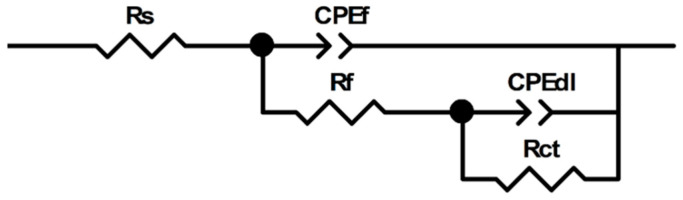
Equivalent circuit model used to fit the impedance data.

**Figure 7 materials-15-04218-f007:**
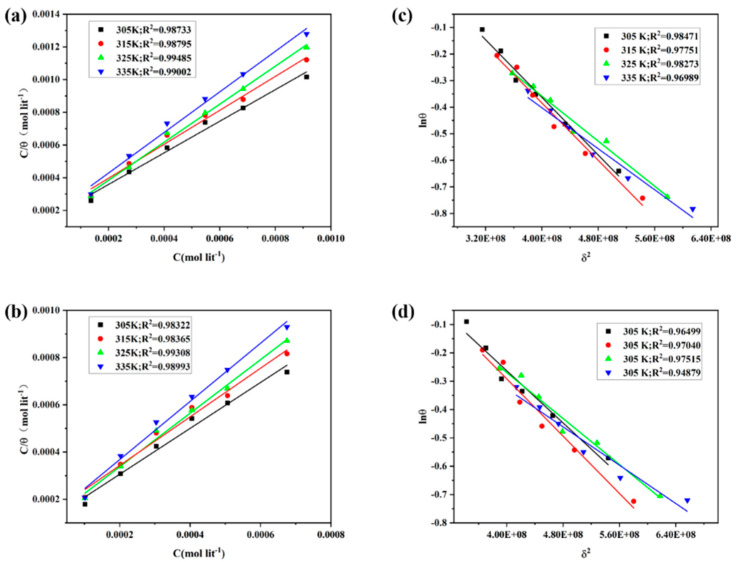
Langmuir isotherm (**a**) MPO, (**b**) MPPO and Dubinin–Radushkevich isotherm (**c**) MPO, (**d**) MPPO plots for N80 steel in 1 M HCl solution containing different concentrations of MPO and MPPO at different temperature.

**Figure 8 materials-15-04218-f008:**
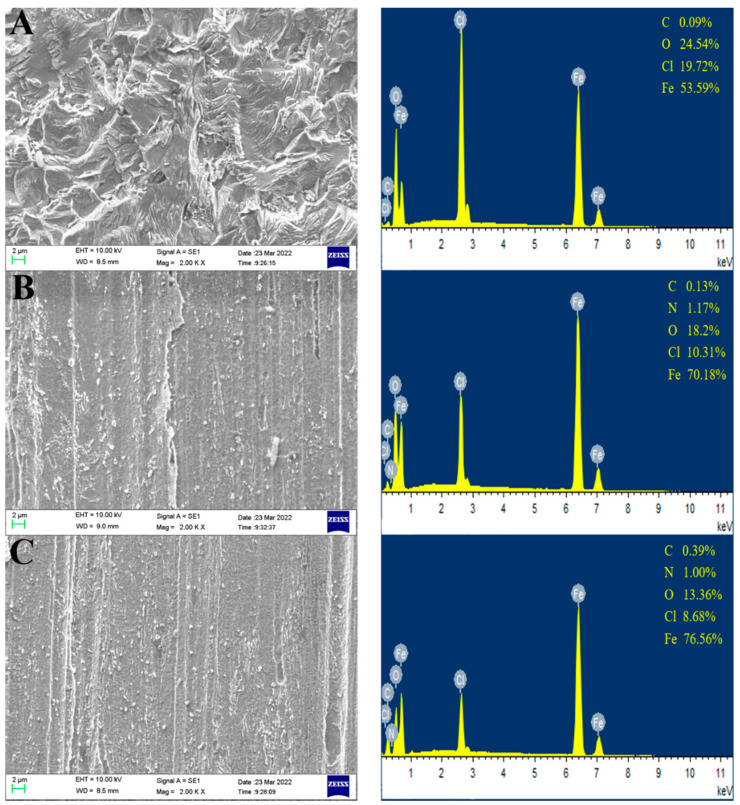
SEM micrographs and their corresponding EDS analysis of N80 steel after 2 h immersion in 1 M HCl solution: (**A**) without inhibitors; (**B**) presence of 300 ppm of MPO; (**C**) presence of 300 ppm of MPPO.

**Figure 9 materials-15-04218-f009:**
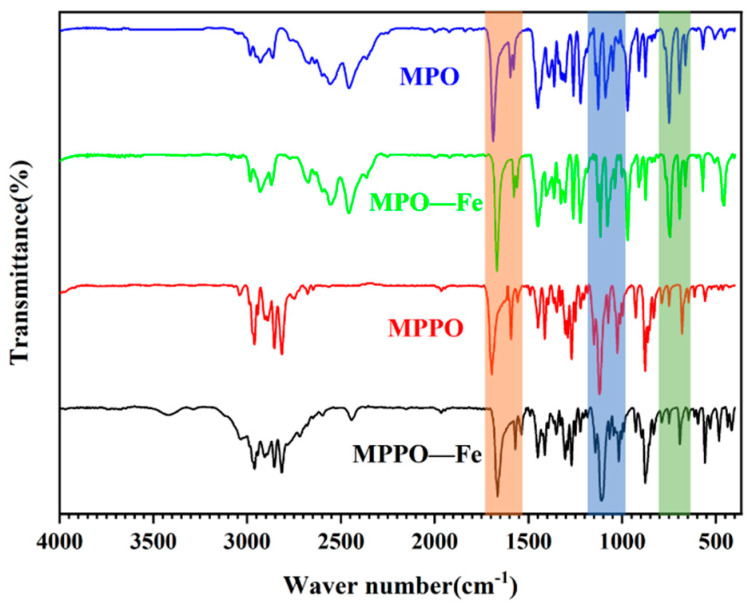
FT-IR adsorption spectrum of free inhibitors (MPO and MPPO) and scrapped samples (MPO-Fe and MPPO-Fe).

**Table 1 materials-15-04218-t001:** Corrosion parameters obtained by weight loss measurements of N80 steel in 1.0 M HCl containing various concentrations of MPO and MPPO at different temperatures.

Temperature (K)	Concentration (ppm)	MPO			MPPO		
		C_R_ (mg cm^−2^h^−1^)	IE (%)	θ	C_R_ (mg cm^−2^h^−1^)	IE (%)	θ
305	0	1.86	-	-	1.86	-	-
	50	0.88	52.7	0.527	0.81	56.5	0.565
	100	0.69	62.9	0.629	0.64	65.6	0.656
	150	0.55	70.4	0.704	0.53	71.5	0.715
	200	0.48	74.2	0.742	0.47	74.7	0.747
	250	0.32	82.8	0.828	0.31	83.3	0.833
	300	0.19	89.8	0.898	0.16	91.4	0.914
	350	0.17	90.7	0.907	0.14	92.5	0.925
315	0	2.31	-	-	2.31	-	-
	50	1.21	47.6	0.476	1.19	48.5	0.485
	100	1.01	56.3	0.563	0.97	58.1	0.581
	150	0.87	62.3	0.623	0.85	63.2	0.632
	200	0.69	70.1	0.701	0.72	68.8	0.688
	250	0.51	77.9	0.779	0.48	79.2	0.792
	300	0.43	81.4	0.814	0.40	82.7	0.827
	350	0.41	82.3	0.823	0.38	83.5	0.835
325	0	3.24	-	-	3.24	-	-
	50	1.69	47.8	0.478	1.64	49.4	0.494
	100	1.33	59.0	0.59	1.31	59.6	0.596
	150	1.26	61.1	0.611	1.23	62.0	0.620
	200	1.01	68.8	0.688	0.97	70.1	0.701
	250	0.89	72.5	0.725	0.79	75.6	0.756
	300	0.77	76.2	0.762	0.73	77.5	0.775
	350	0.75	76.9	0.769	0.70	78.4	0.784
335	0	3.76	-	-	3.76	-	-
	50	1.96	47.9	0.457	1.93	48.7	0.487
	100	1.83	51.3	0.513	1.78	52.7	0.527
	150	1.65	56.1	0.561	1.59	57.7	0.577
	200	1.43	62.0	0.620	1.36	63.8	0.638
	250	1.28	66.2	0.662	1.22	67.6	0.676
	300	1.08	71.3	0.713	1.03	72.6	0.726
	350	1.06	71.8	0.718	1.01	73.1	0.731

**Table 2 materials-15-04218-t002:** Activation parameters for the corrosion of N80 steel in 1M HCl at various concentrations of morpholinyl Mannich base inhibitor.

Inhibitor	Concentration (ppm)	E_a_ (kJ mol^−1^)	A (mg cm^−2^h^−1^)	$∆H_{corr}^{*}(\mathrm{kJ}\;{\mathrm{mol}}^{-1})$	$-∆S_{corr}^{*}(J\;{\mathrm{mol}}^{-1}K^{-1})$
Blank	0	20.84	6.83 × 10^3^	18.18	180.43
MPO	50	23.33	8.88 × 10^3^	20.67	178.25
	100	27.22	3.20 × 10^4^	24.56	167.59
	150	31.23	1.27 × 10^5^	28.58	156.14
	200	31.05	9.86 × 10^5^	28.40	158.23
	250	40.10	2.34 × 10^6^	37.44	131.89
	300	49.47	6.19 × 10^7^	46.81	104.66
MPPO	50	25.46	1.92 × 10^4^	22.75	171.00
	100	29.23	6.58 × 10^4^	26.52	161.75
	150	31.86	1.56 × 10^5^	29.16	154.54
	200	30.23	7.13 × 10^5^	27.52	161.08
	250	39.79	1.97 × 10^6^	37.09	133.48
	300	53.99	3.13 × 10^8^	51.28	91.33

**Table 3 materials-15-04218-t003:** Electrochemical parameters obtained from polarization curves for N80 steel containing different concentrations of MPO and MPPO.

Inhibitor	Concentration (ppm)	I_corr_ (μA cm^−2^)	−Ecorr (mV)	βa (mV)	−βc (mV)	θ	IE (%)
Blank	0	777.90	420	101	139	-	-
MPO	50	172.27	410	83	126	0.7785	77.85
	100	145.73	394	70	125	0.8127	81.27
	150	90.58	420	73	127	0.8836	88.36
	200	84.86	424	69	122	0.8909	89.09
	250	75.88	409	79	129	0.9025	90.25
	300	75.78	405	69	122	0.9026	90.26
MPPO	50	159.79	420	102	115	0.7946	79.46
	100	154.37	390	72	137	0.8016	80.16
	150	132.98	394	91	162	0.8291	82.91
	200	120.52	382	72	168	0.8451	84.51
	250	98.12	398	73	125	0.8739	87.39
	300	74.19	401	68	119	0.9146	90.46

**Table 4 materials-15-04218-t004:** Electrochemical parameters obtained from EIS data for N80 steel in 1M HCl solution containing different concentrations of MPO and MPPO.

Inhibitor	C (ppm)	R_s_ (Ω cm^2^)	R_CT_ (Ω cm^2^)	Y_0, dl_ (F cm^−2^)	n_dl_	C_dl_ (F cm^−2^)	R_f_ (Ω cm^2^)	Y_0,f_ (Ω^−1^ s^n^ cm^−2^)	n_f_	C_f_ (F cm^−2^)	R_p_ (Ω cm^2^)	IE (%)
Blank	0	2.167	28.3	2.73 × 10^−4^	0.944	1.76 × 10^−4^	5.293	8.84 × 10^−7^	0.950	4.62 × 10^−7^	33.59	-
MPO	50	1.680	83.6	2.74 × 10^−4^	0.916	1.35 × 10^−4^	4.746	2.63 × 10^−6^	0.869	4.81 × 10^−7^	88.307	61.97
	100	2.249	136.8	9.49 × 10^−5^	0.88	2.99 × 10^−5^	5.303	6.50 × 10^−7^	0.971	4.47 × 10^−7^	142.143	76.37
	150	1.869	145.53	1.27 × 10^−4^	0.915	5.87 × 10^−5^	4.582	1.04 × 10^−6^	0.952	5.59 × 10^−7^	150.112	77.63
	200	2.218	180.7	1.36 × 10^−4^	0.921	6.73 × 10^−5^	4.951	8.60 × 10^−7^	0.955	4.79 × 10^−7^	185.651	81.91
	250	2.586	278.1	1.68 × 10^−4^	0.794	2.25 × 10^−5^	4.360	3.06 × 10^−7^	1.035	4.82 × 10^−7^	282.490	88.11
	300	1.770	340.2	6.73 × 10^−5^	0.894	2.30 × 10^−5^	4.813	1.71 × 10^−6^	0.912	5.52 × 10^−7^	345.053	90.27
MPPO	50	0.586	112.0	1.30 × 10^−4^	0.861	2.79 × 10^−5^	1.165	1.70 × 10^−7^	1.302	6.07 × 10^−6^	113.195	70.33
	100	0.609	140.0	1.17 × 10^−4^	0.869	2.76 × 10^−5^	2.111	5.78 × 10^−7^	1.176	4.45 × 10^−6^	142.111	76.37
	150	0.610	156.1	1.29 × 10^−4^	0.849	2.40 × 10^−5^	1.080	2.10 × 10^−7^	1.305	7.51 × 10^−6^	157.170	78.63
	200	0.926	230.5	1.61 × 10^−4^	0.822	2.38 × 10^−5^	3.086	3.63 × 10^−7^	1.182	2.99 × 10^−6^	233.576	85.62
	250	1.945	303.3	1.68 × 10^−4^	0.799	2.24 × 10^−5^	2.493	9.48 × 10^−8^	1.249	1.98 × 10^−6^	305.784	89.02
	300	1.922	349.2	1.25 × 10^−4^	0.890	4.43 × 10^−5^	2.829	3.71 × 10^−7^	1.136	1.93 × 10^−6^	352.049	90.46

**Table 5 materials-15-04218-t005:** Thermodynamic parameters for adsorption of inhibitors in 1.0 M HCl on N80 steel surface at different temperatures.

Inhibitor		Temperature (K)	Langmuir Isotherm			Dubinin–Radushkevich Isotherm		
			R^2^	K_ads_ (Lmol^−1^)	$-∆G_{ads}^{0}\;(\mathrm{kJ}\;{\mathrm{mol}}^{-1})$	R^2^	a (mol^2^ J^−2^)	E_ads_ (kJ mol^−1^)
MPO	305		0.987	7352	32.76	0.985	2.71 × 10^9^	13.58
	315		0.988	7042	33.72	0.978	2.70 × 10^9^	13.62
	325		0.995	6756	34.68	0.983	2.12 × 10^9^	15.37
	335		0.990	5494	35.17	0.97	1.93 × 10^9^	16.1
MPPO	305		0.983	11764	33.95	0.965	2.30 × 10^9^	14.73
	315		0.984	9090	34.39	0.97	2.53 × 10^9^	14.07
	325		0.993	8264	35.22	0.975	2.03 × 10^9^	15.7
	335		0.990	7751	36.13	0.949	1.69 × 10^9^	17.21

**Table 6 materials-15-04218-t006:** FT-IR absorption frequencies for the free inhibitors and the adsorbed inhibitors on the N80 steel surface.

Inhibitor MPO	Sample MPO-Fe	Inhibitor MPPO	Sample MPPO-Fe	Tentative Assignment
1687	1668	1695	1664	ν(C=O)
1597	1577	1593	1570	ν(C=C)&ν(C=N) pyridine ring
1089	1083	1026	1018	ν(C-O-C) morpholine ring
1128	1121	1120	1110	ν(C-N-C)
750	756	680	692	C=O deformation
-	569	-	559	ν(Fe-O)
-	457	-	484	ν(Fe-N)

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
