# Peer review of "Newly Synthesized Morpholinyl Mannich Bases as Corrosion Inhibitors for N80 Steel in Acid Environment"

_materials, 2022, doi:10.3390/ma15124218_

Round 1

Reviewer 1 Report

The manuscript ID: materials-1740821: "Effect of morpholinyl Mannich base on corrosion and inhibition of N80 steel in HCl." discusses the synthesis and characterization of two new corrosion inhibitors for N80 steel in 1.0 M HCl solution. The research presented in this manuscript is of interest, but should be considered for publication only after addressing the following issues:

  1. Title (lines 2–3): The title needs to be changed to: “Newly synthetized morpholinyl Mannich bases as corrosion inhibitors for N80 steel in acid environment”.

  1. Abstact (line11) and elsewhere in the manuscript: Once HCl was used after hydrochloric acid (not needed in my opinion) the “hydrochloric acid” should not be used anymore.

  1. Abstract (lines 14–15) and Conclusions (line 377): The authors need to specify to which compound do the 89.8% and 91.4% values belong to. The above-mentioned values are however not the highest IE values obtained for 300 ppm inhibitor (please check the values from polarization and EIS measurements).

  1. Introduction (lines 33–36): Please rephrase these sentences. I doubt the high price (not value) presents a problem in the case of inorganic corrosion inhibitors. Moreover, the authors need to specify which inorganic substances are not allowed to be used, because there are still inorganic compounds in use as corrosion inhibitors.

  1. Introduction (lines 36–38): What do the authors mean with “… may contribute to the lone pair electrons”? The whole sentence makes no sense and need to be rephrased.

  1. Introduction: The authors mention in lines 47–49 that Mannich bases are found to be excellent corrosion inhibitors. However, the authors fail to provide further discussion and references to expand that. They present only three cases when morpholine derivatives have been employed as corrosion inhibitors.

  1. Introduction (lines 60–61): The authors should add MPO as well.

  1. Materials and Methods: The authors must provide for all chemicals and equipment the name of the company and the country.

  1. Materials and Methods (line 91): Was the N80 purchased? If so, give the name and country of the provider. The authors should give the diameter and thickness of the samples (discs). In the manuscript the authors write “oil tube steel discs” which might lead the reader to wrongly believe that the samples were taken from the oil tubes, which I do not believe to be the case here.

  1. Materials and Methods (lines 93–94): Replace “mechanically polished” with “ground”. The authors should mention the range (from the lowest to the highest) of the grinding paper.

  1. Materials and Methods (lines 100–101): What do the authors mean with “atmospheric conditions”? What is the immersion time of the samples in 1.0 M HCl solution during the weight loss measurements?

  1. Materials and Methods (lines 101–102): The following sentence needs rephrasing “The previous experiments were repeated in each case and the results were averaged”.

  1. Materials and Methods (line 114): The authors mention: “… maintaining a constant temperature of 25 °.”. This is not always the case, as the authors tested the inhibitors at different temperatures as well.

  1. Materials and Methods (lines 118–120): Were EIS and polarization performed in a sequence in the same sample (EIS and then polarization) or on different samples? Add “OCP” after “open circuit potential” so you have the abbreviation used later on.

  1. Materials and Methods (lines 123–127): These lines do not belong to the “Electrochemical Measurements”, but to the “Surface analyses” (which I would suggest renaming it to “Surface characterization”).

  1. Materials and Methods (lines 129–130): The authors refer to the samples as “discs”, “plates”, “specimens”. One term (samples) should be used throughout the manuscript. What is the shape of the samples used for surface characterization? Are those the same samples as the ones used for weight loss measurements?

  1. Materials and Methods (line 130): What does PPZs stands for?

  1. Materials and Methods (lines 138–140): The samples for SEM analysis were immersed only for 2 h in the 1.0 M HCl solution with and without inhibitor solution. Meanwhile, for UV-VIS the immersion time was 48 h. Why such a big difference in the exposure time? At what wavelength was the measurement performed?

  1. Materials and Methods (lines 138–140): The authors must prove that the corrosion inhibitors are adsorbed on the surface of the N80 steel (FTIR measurements), since they claim later in the manuscript that a film is formed due to the adsorption. The UV-VIS measurement is worthless in my opinion, as it does not prove that the corrosion inhibitors are on the surface of N80.

  1. Results (lines 145–146): The sentence containing “… the temperature vary of 305-335K square measure listed in table 1” must be rephrased.

  1. Results (lines 148–150): The authors state: “It is observed that all inhibitors showed very high efficiency at the optimum concentration of 300ppm, and there is no remarkable increase in the inhibition efficiency for a higher concentration of inhibitors.”. In order to prove that 300 ppm was the optimum concentration the authors must present results for concentrations higher than 300 ppm, so the readers can see that there is no significant change in the IE. In Table 1 the authors use “η (%)”, while elsewhere use “IE (%)”. The authors should use only IE(%) throughout the manuscript.

  1. Results (lines 150–151): The explanation given is short and not clear. The authors must explain in more details why no further increase of IE was observed after 300 ppm.

  1. Results (lines 158–164): The discussion about the effect of temperature is such a mess and need to be rewritten. First, to which inhibitor do the CR values given at 305 and 335 K, belong to? Second, the authors must mention what happened to the CR with the increase of temperature for the samples immersed in the 1.0 M HCl solution with and without inhibitor addition.

  1. Results (lines 220–222): Figure 4 shows no significant shift in the Ecorr of the MPO inhibited samples with regard to the non-inhibited samples. Meanwhile, a shift in the Ecorr of the MPPO inhibited samples (towards more negative values) is observed.

  1. Results (lines 224–227): The following sentence must be rephrased: “In brief, among the studied inhibitors show better inhibition efficiencies and the efficiencies increase with the concentrations of inhibitors, which is due to the absorbed inhibitors forming a protective film on the steel surface.”.

  1. Results (lines 281–282): The type of EIS response of the inhibited and non-inhibited samples is identical. I do not understand why two different circuits are used to fit the EIS spectra.

  1. Results (line 273): Table 4 is a mess. First, what are the units for the resistances? Second, for the Cdl, Cf and the respective Y values, write in the table on the value and have the 10x in the unit. For example “Cdl (unit, x 10–4)” and the values in the table “1.33; 1.35, 0.299..”. Third, use IE(%).

  1. Results (lines 358–369): In my opinion this is worthless. Even if a complex is formed, it might be formed in the solution (with the Fe2+ from the oxidation process), and not on the surface of the N80 steel.

  1. General comment: The text must undergo serious English editing, from a native English speaking editor, before being reconsidered for publication.

Reviewer 2 Report

Review of paper no. materials-1740821 titled Effect of morpholinyl Mannich base on corrosion and inhibition of N80 steel in HCl by Y. Chen et al.

In this paper the authors studied the corrosion inhibition of N80 steel by two organic compounds – Mannich bases. It is found that the compounds can efficiently decrease the corrosion rate. The paper is sufficiently detailed and employs a large number of experimental techniques (weight loss measurements, potentiodynamic polarization experiments, EIS, SEM/EDS, XPS, UV-vis spectroscopy). It is written in good English. The manuscript is publishable subject to revision.

1.Table 1 provides an overview of the corrosion rates calculated from the weight loss data. However, the raw weight loss data is not presented. Please, include the weight loss curves in the manuscript.

2.Corrosion resistance was studied by potentiodynamic polarization (Fig. 4). Have you measured an open circuit potential (OCP) before the polarization? If so, include it in the manuscript.

3.It is difficult to verify that the Mannich bases were successfully adsorbed on the steel surface from the SEM/EDS results (Fig. 8). The EDS analysis is not suitable for light elements like N or C. Have you considered analyzing the materials by Raman or IR spectroscopy?

4.The paper lacks a comparison with the literature. The results obtained (inhibitor efficiency, corrosion rates) should be compared with previously published works (https://doi.org/10.1016/j.molliq.2021.117957, https://doi.org/10.1108/ACMM-05-2019-2119).

Minor points:

5.Lines 91-92: Is the steel chemical composition given in wt.%? Please, specify.

6.Line 116: The saturated glycerol electrode does not exist. You perhaps meant a saturated CALOMEL electrode. Please, check.

7.The mathematical equations used in the manuscript should be numbered.

Round 2

Reviewer 1 Report

In the revised manuscript materials-1740821 the authors have addressed the vast majority of my concerns. However, in my opinion there are still minor issues to be addressed. 

1. The providers of the chemicals/equipment must be given for each one, or grouped when the provider is the same. Please change this in lines 73 and elsewhere if needed.

2. In your response to my question your write: "at standard atmospheric pressure". How can you maintain the same pressure in all your measurements? This does not make sense.

3. In your response you mention that EIS and polarization was performed in the same sample after OCP. I guess EIS was performed after 1 h of immersion (it is not specified in the manuscript). What about the polarization? After how many hours after immersion were the polarization curves obtained? Did you perform OCP measurements after EIS? Please specify this in the manuscript.

4. I do not see the point in having both Table 1 and Figure 3 or Table 1 and Figure 4. In both cases the table and the figure tell the same story, nothing new.  

5. You have not addressed my point regarding why 300 ppm was considered the optimum concentration. Data should be given about measurements higher than 300 ppm showing no improvement in the CR.

6. You have not reflected my comment regarding the presentation of the data in Table 4. It is not acceptable to have the values as "... E-7", etc. Please refer to my suggestion on my first review report. I find it strange that you use now only one circuit to fit the EIS data, and the fitting results (values obtained) did not change.

Reviewer 2 Report

Authors answered most of my comments. The paper is publishable subject to minor revision.

1.The OCP curves (Fig. 2) are results. As such, they should be moved to the results section. I recommend presenting them together with Tafel polarization curves (section 3.2). The section 3.2 should be renamed to Electrochemical corrosion measurements.
